# Using Photovoice to Explore Determinants of Health among Homeless and Unstably Housed Women

**DOI:** 10.3390/ijerph21020195

**Published:** 2024-02-08

**Authors:** Jessica L. Mackelprang, Janessa M. Graves, Halle M. Schulz

**Affiliations:** 1Department of Psychological Sciences, Swinburne University of Technology, Melbourne 3122, Australia; 2Department of Family Medicine, School of Medicine, University of Washington, Seattle, WA 98195, USA; 3College of Nursing-Spokane, Washington State University, Spokane, WA 99201, USA; halle.schulz@wsu.edu; 4Honors College, Washington State University, Pullman, WA 99164, USA

**Keywords:** homelessness, housing instability, women, Photovoice, social ecological model

## Abstract

The lived experiences of homeless and unstably housed women, including their health-related priorities, are understudied in smaller metropolitan and rural communities. In this study, we partnered with a day center for women who experience homelessness in Spokane, Washington. We used Photovoice, a community-based participatory action research method, to explore the health-related concerns, needs, and behavior of women who are homeless or unstably housed. Participant-generated photographs and group interview data were analyzed using thematic analysis. Three themes were generated: “These are my supports”, “I’m trying to make my health better”, and “[My] choices are very limited”. The themes illustrated individual, interpersonal, community, and societal strengths and vulnerabilities aligned with the social ecological model. Participants demonstrated resourcefulness, creativity, and hope as they strived toward health improvement. Trauma-informed, strengths-based approaches that respect the autonomy of homeless and unstably housed women and that amplify their voices are needed to minimize power imbalances in research, policy, and practice. This includes an imperative for healthcare and social work programs to ready graduates to deliver effective, empathic services by increasing their knowledge of social determinants of health and of the stigma faced by marginalized communities. Moreover, collaborating with these communities when designing, implementing, and evaluating services is critical.

## 1. Introduction

Homelessness is a serious public health concern in the United States (US), including in smaller metropolitan and rural communities. In 2019, an estimated 1309 individuals were homeless in Spokane, Washington; 39% were women [1]. Precariously housed people experience health disparities and unmet health needs [2,3,4,5,6], and homeless women may experience more severe health problems (e.g., chronic illness, chronic mental health conditions) and greater health service utilization than men [7,8]. Gendered aspects of homelessness, including higher rates of violent victimization prior to and during homelessness among women, also influence psychiatric morbidity and wellbeing [9,10,11]. Homeless and unstably housed women experience both “spatial and policy invisibility” [12] (p. 395). Their day-to-day experiences, including their perspectives on health and desired supports, remain understudied, particularly in less-populated communities.

Qualitative methodologies contextualize perspectives and inform the development of interventions [13]. Community-based participatory action research methods, such as Photovoice, bridge the gap between science and practice [14] and discern factors that influence wellbeing among marginalized populations [15,16]. When exploring the experiences of homeless and unstably housed women, strengths-based, trauma-informed approaches are needed. Such approaches reduce the risk of perpetuating historical power imbalances between participants and researchers [17]. Trauma-informed approaches acknowledge the impacts of trauma; create physically and emotionally safe spaces; collaborate with trauma survivors to support self-determination; maintain a strengths-based engagement approach; and are sensitive to cultural, historical, and social factors that influence participants’ lives, including gender [18].

Photovoice is rooted in feminist theory, empowerment education, and documentary photography. It uses photography and dialogue to: (1) document individual and community strengths and vulnerabilities; (2) encourage discourse about health and social issues; and (3) spur change by prompting critical community discourse [19]. Photovoice has been used to explore lived experiences of homelessness and social inequity [20,21,22], to clarify individuals’ priorities [23], and to examine health needs [15,24]. Despite substantive application within homeless communities, few Photovoice studies foreground the health needs of unstably housed women [25,26].

Photovoice is a compelling means of exploring individuals’ portrayal of their lives and of honoring subjective experiences [27]. We used Photovoice to explore health-related concerns, needs, and behaviors of homeless and unstably housed women in a moderately sized Pacific Northwest US city. We applied the social ecological model to highlight individual, interpersonal, community, and societal strengths among these women. Further, we strived to give voice to their lived experiences and to inform suggestions for community organizations, healthcare providers, and educators.

## 2. Materials and Methods

### 2.1. Setting and Procedures

Women’s Hearth provides support for homeless and low-income women (more than 1000 women annually) in the heart of Spokane, Washington, USA. Spokane is situated in Eastern Washington State and had a population of 230,160 in 2022. Guided by four core values (i.e., respect for human dignity, community, growth and wholeness, justice), Women’s Hearth offers activities, referrals, and housing assistance, as well as hygiene support, food, and technology. Among the activities is a weekly photography class that covers composition, light, and framing. Participants were recruited through this class.

Recruitment occurred during an August 2016 photography class (Session 1); homeless and unstably housed women were eligible. Attendees were informed about the voluntary study, which involved photography assignments and group discussions (Sessions 2 and 3). Written informed consent was obtained and demographic information collected. Participants were provided with a digital camera (i.e., Kodak PIXPRO Friendly Zoom FZ41 16 MP) in exchange for depositing a self-selected collateral item at Women’s Hearth. Participants received a bus pass to reimburse transportation costs and a CD containing their project photographs.

During Sessions 1 and 2, the participants were prompted to photograph health-related aspects of their lives (e.g., “What does health mean to you?” “What does it mean to be unhealthy?”). Writing photo captions was optional. During Sessions 2 and 3, participants explored individual and collective meanings of their photographs during 2 h long discussions that were co-facilitated by a photography teacher and a researcher (H.M.S.). Semi-structured prompts (e.g., “Describe what is going on in this photo.” “What made you decide to take this photo?” “What does this photo mean to you?”) explored the stories associated with each photograph.

Discussions were audio-recorded and transcribed verbatim. Participant names were removed prior to data analysis and replaced with pseudonyms to protect their identities. Photograph captions were included in data analysis. Following data collection, participants selected photographs for exhibition at Women’s Hearth. Study procedures were approved by Washington State University’s Institutional Review Board (IRB number 15316).

### 2.2. Data Analysis

This study was approached from an experiential orientation underpinned by a critical realist ontology and contextual epistemology. Thematic analysis was used for data analysis [28]. Data were analyzed using a quasi-inductive approach, wherein we coded transcripts inductively. First, we familiarized ourselves with audio recordings, transcripts, and photographs. We completed initial transcript coding individually, alongside reviewing accompanying photographs. We then met virtually to collate codes, which involved pooling codes, eliminating redundant or idiosyncratic codes, and generating candidate themes. The recursive process of developing themes occurred during research team conversations and while drafting this manuscript, during which we revisited the dataset to refine and name themes.

Partners at Women’s Hearth were invited to provide feedback on the findings. Finally, themes were considered in relation to the social ecological model [29]. This iterative process triangulated the researchers’ perspectives and those of our community partner, producing a more nuanced understanding of the data and ensuring rigor and trustworthiness [30]. Dedoose software (Version 7.0.23, 2016) was used for data management.

### 2.3. Positionality Statement

The researchers comprised two psychology and nursing academics, respectively, and an undergraduate nursing student; all are Caucasian women. Two members lived and/or work in the community wherein the study was conducted. The team has over a decade of experience working with marginalized communities in the US and internationally, though none have lived experience of homelessness. We have partnered with people experiencing homelessness in research and have provided health and mental health services for these communities.

## 3. Findings

Six women participated. Their ages were recorded as ranges: 18–35 (*n* = 1), 46–55 (*n* = 2), and 56–65 (*n* = 3). All participants identified as Caucasian and non-Hispanic. Two reported intermittent or chronic homelessness in the past three years; the others were unstably housed. Four had completed high school, one had less than a high school education, and the other had completed community college. Three participants were unemployed, one reported being disabled, and two declined to report employment status.

We constructed three themes that encapsulated behavioral and structural factors that influenced participants’ physical and emotional health: “These are my supports”, “I’m trying to make my health better”, and “[My] choices are very limited”. Within each, we identified the barriers and facilitators to health consistent with the levels outlined in the social ecological model (Table 1). Hereafter, the participants’ words and photographs enliven each theme, with information related to the social ecological model woven throughout.

### 3.1. Theme 1: “These Are My Supports”

Participants emphasized the importance of instrumental and emotional support, which were facilitators for maintaining or improving physical and mental health. Instrumental support from community- (e.g., food bank, volunteer organizations) and government-funded programs (e.g., food stamps, social security) was relied upon for food (e.g., groceries), hygiene products, and medications. For instance, Nina relied on homeless shelters for showers and a safe place to sleep. Shelters were also an important source of personal care supplies for her, “*If I’m out of deodorant I go to where I can get the deodorant because [shelter] got it sometimes*”.

Instrumental and emotional support interwove among the women at the center who kept one another “*emotionally sane*”. Support from homelessness sector workers, faith communities and, for two participants, family were valued. Referring to women’s center staff who helped her apply for housing, Nina stated, “*No matter what the problem is you can talk to them*”. She transitioned from rough sleeping to shared accommodation during the study, with encouragement from staff and fellow participants, and she speculated that without them, she would not have obtained housing. One of Joan’s photographs—two women holding hands—exemplified this theme, illustrating the significance of instrumental and emotional support (Figure 1).

Participants’ stories revealed the importance of physical items as sources of comfort or attachment. For instance, Nina and Joan felt attached to particular dishes at the women’s center, which symbolized the sense of home they experienced there. Patti found “*joy*” in nature and considered church “*part of [her] support and guidance*”. Nina had a stuffed bear (“*emotion animal*”) and shared, “*When I’m so emotion[ally] upset or depressed, I hug onto it, and she takes it away from me*”. Four participants mentioned or photographed a walker. Although assistive devices were cumbersome (e.g., on public transit), participants viewed them as life-enhancing. Walkers provided physical support, enabling participants to ambulate safely (e.g., “*It gives me some security and it always gives me a place to sit down… like [when] my legs start to giving*”—Patti) and to “*help other people [when they’ve fallen]*” (Cheryl). Walkers helped participants cope with health conditions that made mobility difficult and facilitated exercise (e.g., walks), particularly when tired. Patti’s walker enabled her to eat healthier, as she could spend more time cooking. She stated, “*I’ll take my walker over by the stove and sit in it*”.

### 3.2. Theme 2: “I’m Trying to Make My Health Better”

Participants reflected on behavior they considered healthy or unhealthy; explored ways they wanted to improve their health; and discussed their actions toward bettering wellbeing, including motivations for change. Group rapport enabled participants to call out risky or unhealthy behavior (e.g., smoking) and to tease one another about the ways in which they were or were not progressing toward professed goals. They agreed “*change is always scary*”.

Nutrition was a ubiquitous health improvement target, and efforts to improve diet was an individual-level health facilitator. Images of food spurred conversation about whether they eat healthily or not, grocery costs, and dietary restrictions associated with health conditions (e.g., diabetes). Patti stated, “*I try to eat more fresh fruits and vegetables and drink more water*”. She continued, “*I’m trying to make my health better… I watch what I eat*”. Sweets were photographed frequently, and sugar was lamented for making food “*unhealthy*” or “*bad*”. The sentiment that “*sweets also kind of make your life feel special*” (Patti) highlighted that emotional factors influence dietary decisions. Illustrating this tension, Joan and Grace described having a “*stash*” of sweets, despite discomfort due to dental problems that was worsened by sweets.

Striving to increase physical activity featured frequently in discussions, and limited economic resources was an obvious barrier to some activities. Notwithstanding this obstacle, participants made efforts to increase movement and sought physical activity opportunities through community programs (e.g., swimming). Walking was participants’ main form of exercise and primary mode of transportation, alongside public transit: 

*When I’m at home, [I walk] from home to the bus stop in the morning, and walk around here and the gas station, and walk to the bus stop again and get on the bus, and do a little at the plaza, then walk to my apartment building and up to my second-floor apartment*.(Grace)

Cheryl asserted, “*[Walking] helps my blood sugar level go down*”, and described benefits to her overall health. Social connectedness also facilitated better health by helping Joan to establish a walking routine and giving her something to look forward to. She stated, “*I walk with my aunt at least once a week*”. Other activities included Wii Bowling and swimming. Referring to a community pool photograph, Patti shared, “*This is where I get some of my exercise and help work on lifting my legs… I can do a lot of stuff in the pool—pretend I’m a ballerina*”.

Physical health status was a barrier to increasing physical activity to desired levels. Patti photographed her edematous leg (“*unhealthy leg*”), which represented her desire to lose weight and to get “*shape*” in her legs, a goal complicated by medical comorbidities and mobility limitations that impeded activities of daily living (e.g., getting in or out of a bathtub independently). For Nina, health influenced sleeping arrangements, specifically which bed the shelter allocated to her (Figure 2).

Beyond concrete activities to improve physical health, participants described leisure activities that kept them “*mentally healthy*” (Joan). Cheryl maintained a meditation practice, “*I always try to meditate every Monday… it helps me be relaxed*”. Joan and Patti found pleasure in crocheting, which, for Patti, was also a means of “*leaving a heritage*” for grandchildren. Other efforts to improve wellbeing were related to hygiene and housing. Maintaining good hygiene was a priority for participants, who photographed shampoo, scrub brushes, and soaps. Hygiene influenced Nina’s sense of self-worth, “*It makes me feel cleaner and more excited and happy for myself that I’m actually taking care of myself instead of just sitting around and letting other people say, ‘You need to take a shower*”.

Accommodation impacted participants’ wellbeing. Grace shared a photograph of her bed, which was covered with her belongings, leaving her nowhere to sleep except the sofa, which hurt her back. In a group discussion, she explored her motivation to clean her bed, namely comfort. In the subsequent session, she shared a photo of her clean bed, expressing pride that she had taken action and exemplifying the health-promoting impact of within-group support. Reflecting on transitioning into housing during the study, Nina observed that her photos documented progress toward stabilization, “*You would know that I was homeless… those pictures I took, they basically show the life story how I was living day-by-day. But now, they can look [at] the picture and say, ’Oh she did it. She’s doing good’*”.

Lastly, participants’ efforts to manage chronic conditions involved seeking healthcare; four women mentioned providers (e.g., nutritionist, podiatrist). Medications to manage chronic illnesses featured in numerous photos (e.g., insulin needle, pill cutter; Figure 3) and were presented as examples of participants’ efforts to tend to their health. As an example, diabetes impacted Cheryl’s nutritional and medication needs, “*I take my meds to stay alive. I try to watch what I eat and drink. I test myself three times a day*”. Joan was focused on dental health and “*brushing [her] teeth a lot more*”. Likewise, Patti flossed nightly, heeding warnings that she might lose her teeth if she did not (“*I love my teeth*”).

Although participants wanted to improve their health, some described “*unhealthy*” or health risk behaviors but no interest in altering their behavior. The most hazardous of these was described by Cheryl, who divulged that she smokes cigarettes in bed and had burned her chest and the bed. Fellow participants expressed safety concerns, but she indicated she had been bed smoking for a “*long time*”, was “*too old to change*”, and had no intention of stopping.

### 3.3. Theme 3: “[My] Choices Are Very Limited”

Although participants were grateful for support from community agencies and other services, they also expressed frustration at being offered few options on health-related matters. Barriers were particularly apparent at community and societal levels, where participants experienced the least power to enact change.

Food cost was cited as a barrier to improving health and influenced participants’ food purchases, which were often economical but not nutritious (Figure 4). The expense of fresh fruits and vegetables was mentioned frequently. Nina explained, “*[Fresh] fruits and vegetables are more expensive sometimes than in a can*”. Grace pointed out that even for people who have money, healthy choices were limited in nearby shops (e.g., gas stations, food bank). Taste was also an issue. Joan indicated that eating healthy makes sense, but “*you want something that tastes good, too, and you see a lot of those like health food stuff is not that tasty*”. Participants agreed that if they had more money, they would buy more fresh foods.

In addition to affecting the choice of what participants ate, money—or a lack thereof—constrained how much they ate. Presenting a photograph of soda and hard-boiled eggs, Nina said, “*That’s all I had to eat that day cause at that time I didn’t have that much to spend*”. Purchasing healthy foods was prioritized below competing needs. For instance, Nina was saving money to rent an apartment, so she opted to purchase cheap food that unfortunately had low nutritional value.

Limited options extended to accommodation, which was allocated to participants, rather than chosen by them. In shared shelter quarters, Nina slept in a bathing suit. The only available pajamas were flannel, which were too hot in warmer months. The lack of privacy frustrated Joan, who reported that roommates sometimes brought people over before she woke up. She hung blankets to shield her bedspace for “*some semblance of privacy*”.

At the societal level, interfacing with the healthcare system was considered integral to wellbeing, but participants speculated that their care was adversely influenced by housing instability. Nina opined that when homeless, “*You go tell your doctor what’s wrong with you and they’ll just turn around and do every test and put you on more pills*”, an observation Patti asserted was because “*they are in cahoots with the pharmacists*”. Participants’ concern about being prescribed medications without considering alternative interventions was summarized by Joan, who stated, “*The problem with pills is that they may solve some problems, but they open you up to more, so that you have to…take more pills to solve the problems*”. These issues, along with last-minute cancellations by healthcare providers and long wait times for prescription refills, undermined participants’ trust in the healthcare system.

## 4. Discussion

We used Photovoice to explore to health concerns, needs, and behavior of homeless and low-income women. The themes tell stories of strength and vulnerability, both of the women participants and of their community. The findings also point to symbolic forms of structural violence that perpetuate disadvantage and affect their health and wellbeing [31]. Health barriers and facilitators paralleled McLeroy et al.’s [29] social ecological model (Table 1), highlighting that intervention points at the individual, interpersonal, community, and societal levels. While some individual- and interpersonal-level factors (e.g., ambivalence about modifying unsafe behavior) impeded participants’ progress toward health goals, most were community- and societal-level barriers.

Participants evinced some knowledge of nutrition; however, for people experiencing food insecurity, knowledge is not associated with better outcomes [32]. The most significant barriers to improving diet—a ubiquitous goal—were the prohibitive cost of nutritious food and limited options. The perception that healthy foods are costly has been documented [33,34]. Participants described sourcing foods from various places to cobble together nutritious meals, exemplifying their resourcefulness and highlighting a community issue: a lack of access to healthy food. The rates of diabetes among people who experience homelessness rival the general population [35], and the lack of healthy foods hinders disease management [36]. Indeed, Kern et al. [37] found that a higher proportion of unhealthy food in local stores is associated with higher insulin resistance. Expanding the inventory of healthy options at community-based services (e.g., food banks, shelters) would be helpful. Individual-level obstacles to healthy eating were evident alongside community- and societal-level barriers. For example, food insecurity may lead to stashing (e.g., candy) [38], as participants described. Clinical and community partnerships to address food insecurity and its adverse health impacts have been described, but further program evaluation is needed [39].

Participants valued community organizations that provided facilities for self-care (e.g., showers), healthy activities (e.g., swimming), and housing support. They derived instrumental and emotional support from agency workers, describing a sense of safety consistent with trauma-informed care, and from other service-accessing women. Reid and colleagues [40] described the value of cultivating trauma-informed, women-only spaces to support women experiencing homelessness, consistent with this study’s setting. Participation in shelter-based activities is associated with wellbeing improvements [41], and social support may motivate physical activity [42]. The success of the Women’s Hearth photography class is an example of the value of group-based activities (e.g., cooking classes, group exercise). Participants’ photos continue to be exhibited to educate health professions students, community members, and elected officials on the lived experiences of precariously housed women. The images illustrate their needs and demonstrate ways in which supported activities meaningfully influence their lives.

Out of necessity, the lives of precariously housed women often entwine with publicly funded services. This study highlighted the integral role community services play in promoting self-efficacy, fostering connection, and nurturing hope. It also revealed that some encounters, such as health care visits, feel dismissive and leave women feeling mistrustful, dehumanized, and stripped of dignity [43,44]. These interactions might be a barrier to subsequent help seeking [45].

### 4.1. Strengths and Limitations

Although our sample size was modest (albeit similar to other Photovoice studies, e.g., [46,47]), multiple rounds of photography and discussion yielded rich data. Having more than one session during which participants shared their photographs and discussed the meanings behind them enabled greater depth in their dialogue regarding health-related concerns, needs, and behaviors. Participants selected the photographs to be exhibited, and partners at Women’s Hearth provided feedback on a draft of this manuscript; however, data analysis was conducted by the research team. The participatory nature of these latter project stages could have been more robust. It warrants mention that this study was conducted in 2016. Although these data are now several years old, homelessness has continued to increase nationally, and the importance of ascertaining and addressing the concerns of people affected by homelessness and housing instability is as pressing as ever. Finally, the purposive demographic of women we recruited was not representative of Spokane’s unstably housed population. However, generalizability was not the goal. These findings may have transferability [48] for women who are homeless or precariously housed in other settings. A study strength was that it was borne of a collaborative partnership between a social support organization, Women’s Hearth, and Washington State University. While established relationships may hinder sharing in some circumstances, it is likely that existing rapport fostered a safe space that facilitated participant candor and discouraged attrition.

### 4.2. Implications for Policy and Practice

Collaborating with homeless and unstably housed women when designing, implementing, and evaluating support services is critical. Decisions regarding activity programming and how activities are integrated into health and social services should be localized [42] and driven by service users to ensure their needs and preferences are prioritized [49].

Training institutions have a responsibility to ready health, mental health, and social service trainees to provide effective, empathic care for impoverished clients upon graduation [50]. This involves educating trainees on the stigma these individuals experience [51] and ensuring they are well positioned to advocate policy changes that serve these populations. As an example, integration of a health equity curriculum has been shown to improve medical students’ knowledge of social determinants of health [52]. Training on applying trauma-informed care principles should be integral to such curricula in social work and other disciplines [53].

Strengths-based, trauma-informed research approaches [18], such as Photovoice, minimize power imbalances between researchers and homeless and unstably housed women, respect their autonomy, and amplify their voices to advocate their needs [54]. Such practices should be standard, given the gendered vulnerability to trauma exposure evident among women who experience homelessness [11]. As an advocacy-driven research approach that elucidates individuals’ priorities, needs, and motivations for behavior change, photographs could be used to prompt exploration of discrepancies between individuals’ goals and behaviors (e.g., motivational interviewing; [55]). Photovoice or other participatory action research methods (e.g., mobile phone diary; [56]) could also be used in healthcare delivery or to evaluate community-led interventions [57].

## 5. Conclusions

Photovoice enables individuals experiencing homelessness and housing instability to demonstrate their resilience and vulnerabilities using photographs and stories, to generate ideas to address the challenges they face, and to influence the support they receive [58]. In this study, Photovoice functioned as a *photovention* and as a *community assessment* [59]. Women utilized photography to critically consider their health and that of other women in their community. Alongside considering individual strengths and vulnerabilities, these findings highlight structural factors that perpetuate social inequalities and may compromise health among low-income women. Themes were centered on the importance of instrumental and emotional support, the challenge of having limited choices, and the resourcefulness and creativity that precariously housed women demonstrate as they strive to improve their health.

## Figures and Tables

**Figure 1 ijerph-21-00195-f001:**
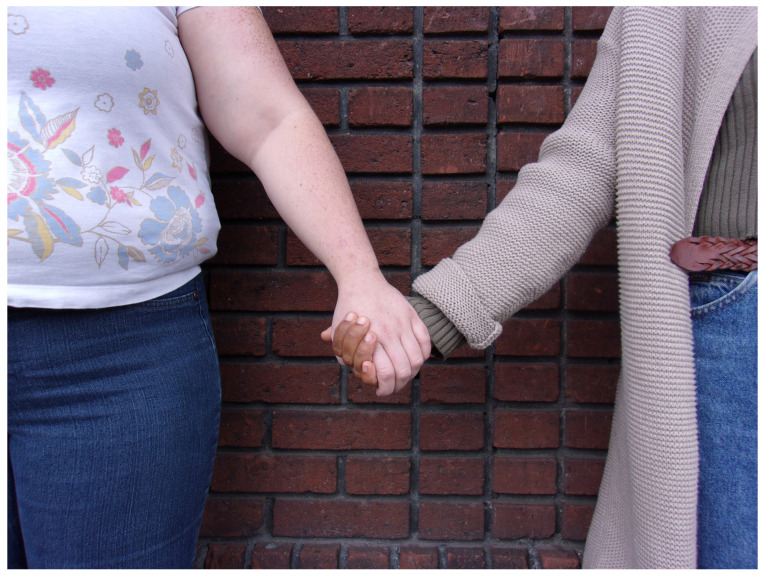
“*One of the things that help me stay emotionally healthy is all the good friends. They help me sort through all the things that make me upset or irritable. I have a lot of good friends that are positive influences on me*”.—Nina. Photo by Joan.

**Figure 2 ijerph-21-00195-f002:**
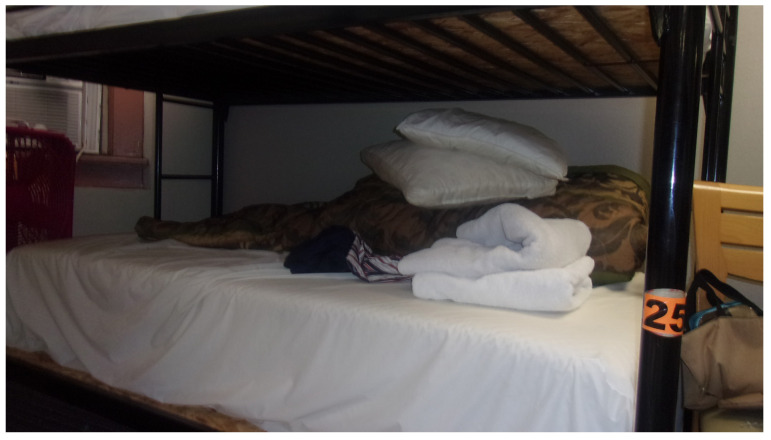
“*My bed [when I was homeless] was always the bottom bunk [at the shelter] because I got some health problems where I cannot climb the top bunk*”.—Nina. Photo by Nina.

**Figure 3 ijerph-21-00195-f003:**
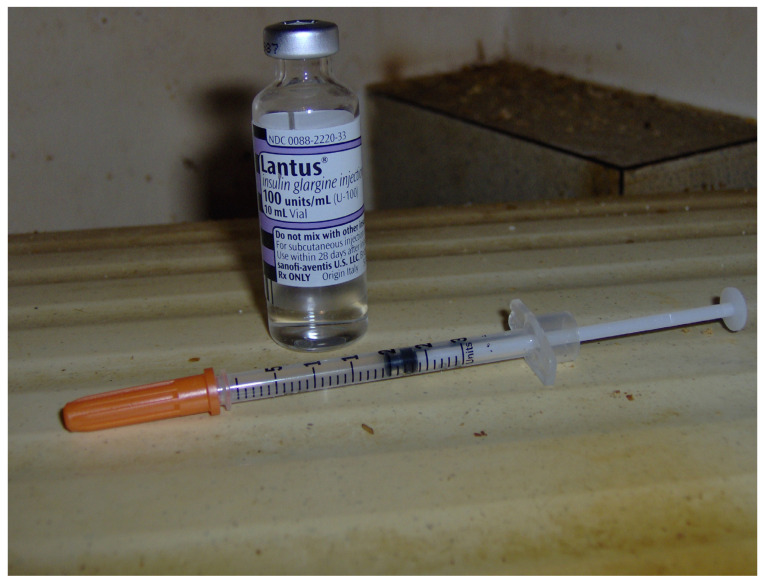
“*I take my shot every night before I go to bed. I got my meds ready for the next day*”.—Cheryl. Photo by Cheryl.

**Figure 4 ijerph-21-00195-f004:**
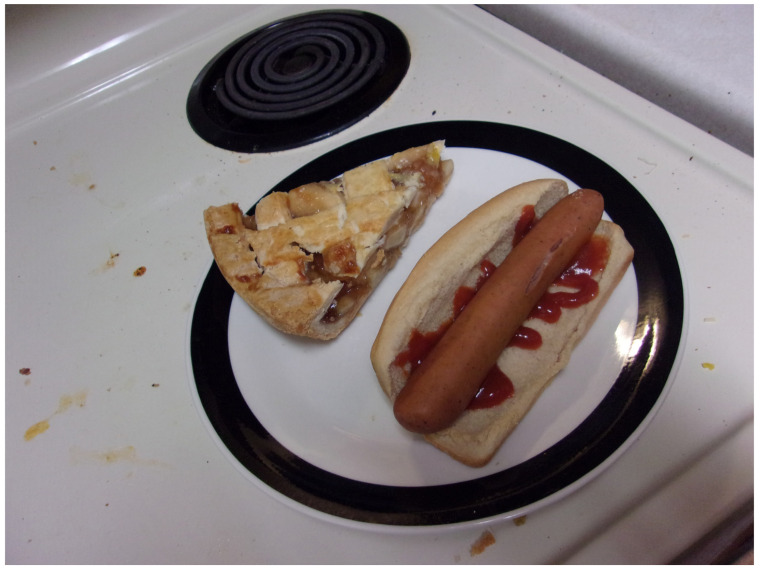
“*Dinner*”—Joan. Photo by Joan.

**Table 1 ijerph-21-00195-t001:** Barriers and facilitators to health in relation to the social ecological model.

SEM Level	Facilitators	Barriers
Individual	Motivated to feel good about themselves (e.g., good hygiene) Keen to improve personal health Basic knowledge of nutrition Access to mobility aids (e.g., walkers) Proactive in seeking healthcare	Pre-existing medical conditions that restrict mobility or exercise Ambivalence about modifying unsafe or unhealthy behaviors (e.g., smoking in bed) Poverty
Interpersonal	Emotional support from workers at homelessness services Emotional support from other women Others’ willingness to engage in joint exercise (e.g., weekly walking)	Limited social support (e.g., family support)
Community	Community organizations that provide practical resources (e.g., food bank) Access to facilities critical to health maintenance (e.g., showers) Access to community-based activities (e.g., swimming pool) Access to public transit	Limited healthy options at food bank and local stores (e.g., gas stations)
Societal	Not applicable	Healthy food (e.g., fresh produce) is more expensive than less nutritious options Mistrust of healthcare providers

Note. SEM = social ecological model.

## Data Availability

The participants in this study did not give written consent for their full data to be shared publicly. Relevant data are contained within the article.

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
