# Peer review of "Using Photovoice to Explore Determinants of Health among Homeless and Unstably Housed Women"

_ijerph, 2024, doi:10.3390/ijerph21020195_

Round 1

Reviewer 1 Report

Comments and Suggestions for Authors

I reviewed the article entitled" Using Photovoice to Explore Determinants of Health among Homeless and Unstably Housed Women".

The authors should be commended for preparing an outstanding, well-written article that was a pleasure to read on a topic that has been understudied.  

The introduction thoughtfully presents the need and thematic framework that guided this qualitative study. 

The methods carefully outlines the specific steps and the fact that the photovoice research project was embedded with recruitment among women in a photography class. All human subjects protection and approvals were applied.

The data analysis and positionality statements are clear and provides important context.

The findings, conclusion and discussions are clearly outlined and presented. 

The key limitation of the study is the small sample size. As the authors pointed out, that is sometimes the case in studies using this type of methodology. However, a few additional comments may reassure the authors of the stability of the findings and conclusions.

Another issue not addressed by the authors is the age of the data. The data collection took place in 2016 and as such the data is almost 8 years old. It may not have have any impact on this topic, the findings and conclusions. However, it should be addressed in the limitations.

In the results, the quotes from the participants were presented with quotation marks. It would be easier for the reader if they were also italicized.

 Again, this article was a pleasure to read and the presentation of the findings and conclusions quite compelling.

The article, given its thematic focus on the social determinants of health will be of great interest to a broad readership and audience focus on improving health conditions for vulnerable women. 

Reviewer 2 Report

Comments and Suggestions for Authors

Amazing manuscript. I enjoyed reading this beautifully written description of this thoughtful study.  The qualitative methods are clearly described. The photos are helpful and thoughtfully explained.

Areas for improvement:

1. The introduction is beautifully written. However, there are some sentences that are rather long with complex structures. Splitting those sentences into shorter sentences may make it easier to read.

2. You mention that the participants had a chance to review the themes/interpretation. Can you clarify how many of the participants engaged in that process?

Reviewer 3 Report

Comments and Suggestions for Authors

This is an excellent manuscript on the experiences of homeless and unstably housed women. At least, 50% of the study participants were post menopausal and all Caucasians. I wondered if you could comment on the neighborhood. Were there other equity deserving groups in the catchment area. Your findings is not going to be generalizable for all homeless or unstably housed women. Similar, homogeneity is highlighted in the research group.

The themes were robust and comprehensive, I did not see a theme of safety or the trauma of contagious diseases. These team may be seen in other diverse shelters. Thank you
